# The Effect of Botulinum Neurotoxin-A (BoNT-A) on Muscle Strength in Adult-Onset Neurological Conditions with Focal Muscle Spasticity: A Systematic Review

**DOI:** 10.3390/toxins16080347

**Published:** 2024-08-08

**Authors:** Renée Gill, Megan Banky, Zonghan Yang, Pablo Medina Mena, Chi Ching Angie Woo, Adam Bryant, John Olver, Elizabeth Moore, Gavin Williams

**Affiliations:** 1Department of Physiotherapy, Epworth Rehabilitation Epworth Healthcare Richmond, Melbourne 3121, Australia; megan.banky@epworth.org.au (M.B.); pab.physio@gmail.com (P.M.M.); angiewoo820@gmail.com (C.C.A.W.); john.olver@epworth.org.au (J.O.); elizabeth.moore@epworth.org.au (E.M.); gavin.williams@epworth.org.au (G.W.); 2School of Physiotherapy, The University of Melbourne, Parkville, Melbourne 3000, Australiaadamlbryant@unimelb.edu.au (A.B.)

**Keywords:** botulinum toxin, muscle strength, upper limb, lower limb, muscle spasticity, Medical Research Council scale, dynamometer

## Abstract

Botulinum neurotoxin-A (BoNT-A) injections are effective for focal spasticity. However, the impact on muscle strength is not established. This study aimed to investigate the effect of BoNT-A injections on muscle strength in adult neurological conditions. Studies were included if they were Randomised Controlled Trials (RCTs), non-RCTs, or cohort studies (*n* ≥ 10) involving participants ≥18 years old receiving BoNT-A injection for spasticity in their upper and/or lower limbs. Eight databases (CINAHL, Cochrane, EMBASE, Google Scholar, Medline, PEDro, Pubmed, Web of Science) were searched in March 2024. The methodology followed Preferred Reporting Items for Systematic Reviews and Meta-Analyses (PRISMA) guidelines and was registered in the Prospective Register of Systematic Reviews (PROSPERO: CRD42022315241). Quality was assessed using the modified Downs and Black checklist and the PEDro scale. Pre-/post-injection agonist, antagonist, and global strength outcomes at short-, medium-, and long-term time points were extracted for analysis. Following duplicate removal, 8536 studies were identified; 54 met the inclusion criteria (3176 participants) and were rated as fair-quality. Twenty studies were analysed as they reported muscle strength specific to the muscle injected. No change in agonist strength after BoNT-A injection was reported in 74% of the results. Most studies’ outcomes were within six weeks post-injection, with few long-term results (i.e., >three months). Overall, the impact of BoNT-A on muscle strength remains inconclusive.

## 1. Introduction

Focal muscle spasticity is common in adult-onset neurological conditions such as stroke, traumatic brain injury, multiple sclerosis, and spinal cord injury [1,2,3,4]. It is a major contributor to disability and the economic burden of disease [5,6,7]. Currently, there is no consensus on the definition of spasticity, but it is characterised as a positive component of the upper motor neuron syndrome (UMNS), resulting in muscle resistance to passive stretch [8,9,10,11,12]. Focal or localised muscle spasticity may cause bodily dysfunction such as pain, contracture or reduced skin integrity and has been shown to negatively impact mobility, upper limb function, personal-care, and quality of life [13,14,15].

Botulinum neurotoxin-A (BoNT-A) intramuscular injection is an effective treatment for focal muscle spasticity [7,15,16,17]. Botulinum neurotoxin-A inhibits the release of acetylcholine neurotransmitters at the neuromuscular junction, reducing nerve impulses to the target muscle [18]. This results in a diminished spastic response to passive movement and reduced involuntary muscle activation [19]. Botulinum neurotoxin-A is effective after a few days, with maximal benefit at four to six weeks post-injection [20]. The BoNT-A effect wears off after approximately 12 weeks, when spasticity may return to a similar or lesser extent [21]. The initial 12 weeks following injection is often referred to as a ‘window of opportunity’ for therapy to improve joint range of motion, muscular strength, and function [22]. However, related studies and systematic reviews have found insufficient evidence of improved function associated with reduced spasticity in the upper and lower limbs [23,24].

While BoNT-A has been shown to reduce spasticity [14,23,25], evidence suggests it may simultaneously induce muscle weakness [11,12]. Animal studies have found that BoNT-A injection causes skeletal muscle dysfunction, including atrophy, weakness, damage to the contractile tissue, and central changes to the modulation of motor neuron excitability [26,27,28]. This may be problematic because emerging literature suggests that muscle weakness and other negative features of the UMNS have a greater impact on function and are associated with increased disability and burden of disease when compared to the positive features [29,30]. Functional gains may require a simultaneous reduction in spasticity and improvements in muscle strength in the limb [31]. However, the impact of BoNT-A on muscular strength remains unclear [17,20,32,33].

Recently, some literature has shown instances of BoNT-A injections causing adverse events, including unintentional weakness of adjacent muscles, contralateral limb weakness, and systemic weakness [34,35,36,37,38,39,40,41,42]. Typically, these reports have low-quality evidence and infrequently use standardised clinical measures for strength [37,40,41,42]. To date, no comprehensive analysis has been undertaken to assess the direct impact of BoNT-A on the muscle strength of both the injected and opposing muscles [17,20,32,33]. Clarifying whether BoNT-A causes muscle weakness is important to aid clinical decisions for injected muscles to ensure efficient synergy of the agonist and antagonist muscles and subsequently avoid possible loss of function. Therefore, the primary aim of this systematic review was to investigate the effect of BoNT-A injections on upper and lower limb muscle strength in adults with neurological conditions.

## 2. Results

### 2.1. Included Studies

Figure 1 presents the flow diagram of study identification to obtain articles for inclusion in the review according to the PRISMA (Preferred Reporting Items for Systematic Reviews and Meta-Analyses) guidelines [43]. A total of 54 articles, comprising 22 RCTs [4,5,32,44,45,46,47,48,49,50,51,52,53,54,55,56,57,58,59,60,61,62] and 32 non-RCTs [19,30,33,63,64,65,66,67,68,69,70,71,72,73,74,75,76,77,78,79,80,81,82,83,84,85,86,87,88,89,90,91], met the inclusion criteria. Among the 22 RCTs, 10 were double-blinded [4,5,48,50,51,55,56,57,61,62], 11 were single-blinded [32,44,45,46,47,49,52,53,54,58,59], and one was unblinded [60].

### 2.2. Study Characteristics

Table 1 (upper limb) and Table 2 (lower limb) summarise the characteristics of the included studies. The 54 studies included 3176 participants assessed before and after BoNT-A injections. Sample sizes ranged from 10 to 333 participants, with a median of 25 participants.

Thirty-eight (70%) studies investigated stroke, ten (19%) studies assessed a mixed cohort (predominantly comprising stroke, traumatic brain injury and cerebral palsy), four (7%) studies investigated Hereditary Spastic Paraplegia and two (4%) studies investigated incomplete spinal cord injury.

### 2.3. Strength Outcomes

Most of the included studies examined upper limb strength outcomes. Thirty-four (63%) studies examined strength outcomes in the upper limb [5,30,44,49,50,52,53,55,57,58,59,60,61,62,63,64,65,66,67,68,69,72,75,77,78,79,82,84,86,87,88,89,90,91], sixteen (30%) studies examined strength outcomes in the lower limb [4,19,32,33,45,46,47,48,51,54,56,70,76,81,83,85], and four (7%) studies reported data for both the upper and lower limbs [71,73,74,80].

Muscle agonists, antagonists, and global muscle strength outcomes were measured across the included studies. Nine (17%) studies reported the strength outcomes of the agonists [19,48,55,66,68,69,71,73,86], six (11%) studies reported strength outcomes for the antagonists [45,47,51,54,83,85], and eighteen (33%) studies reported global strength outcomes [44,52,53,58,59,62,63,64,65,72,76,77,79,80,87,88,89,91]. Eighteen (33%) studies reported a combination of agonist, antagonist, and/or global strength outcomes [5,30,32,33,46,49,50,56,57,61,67,70,75,78,81,82,84,90]. The remaining three (6%) studies did not explicitly specify which muscles or muscle groups were assessed for strength [4,60,74].

#### 2.3.1. Strength Outcome Measurement

Three main types of outcome measures for strength were used in the included studies. Twenty-six (48%) studies used clinical strength outcome measures (e.g., Medical Research Council; MRC), eight (15%) studies used instrumented strength measures (e.g., hand-held dynamometry), and nine (17%) used laboratory-based strength measures (e.g., isokinetic dynamometry) (Table 1 and Table 2). The remaining eleven (20%) studies utilised a combination of these types of strength outcome measures.

The MRC scale was the most commonly used clinical measure, with 27 (50%) studies using it to assess strength [4,5,32,45,46,48,49,51,54,56,60,67,69,70,71,73,74,75,78,80,81,84,85,86,90,91,92]. Ten (19%) studies used the Motricity Index (MI), which incorporates the MRC scale to establish a combined score for upper and lower limb/s strength [32,44,58,59,72,76,77,79,80,91].

Instrumented strength measures included grip and pinch squeeze tests utilising dynamometers [5,52,53,58,59,61,67,72,75,87], pinch gauges [57,67,88], servo meters [61], strain gauges [78], digital force gauges [55,89], and examiner-held dynamometers [50,55]. Four studies stated they used a dynamometer but did not specify the brand or type of device used [58,59,61,87].

Ten (19%) studies assessed strength using laboratory-based measures such as isokinetic dynamometry [19,32,33], force sensors [68,69], dynamic computerised dynamometry [63,64,65], or a force transducer [30,70].

#### 2.3.2. Strength Terminology and Units

Differing terminology was used to define strength, including maximal voluntary contraction (MVC) [19,30,32,33,68,69,70] and maximal voluntary power (MVP)(kg) [50]. When using instrumented or laboratory-based devices, the type of strength measure reported included isometric (*n* = 25) [5,30,33,50,52,53,55,57,58,59,61,62,63,64,65,67,68,69,70,72,75,78,87,88,89] or concentric (*n* = 1) [33]. Strength was measured in Newtons (N) (*n* = 4) [5,30,61,78], Newton meters (Nm) (*n* = 7) [19,32,33,55,61,68,69], kilograms (kg) (*n* = 15) [30,50,52,53,57,58,59,62,63,64,65,67,70,72,75], or scored on an ordinal scale (i.e., MRC or MI) (*n* = 36) [4,5,32,44,45,46,47,48,49,51,54,56,57,58,59,60,66,67,70,71,72,73,74,75,76,77,79,80,81,82,83,84,85,86,90,91]. Two studies did not specify the units used to measure strength [87,89].

### 2.4. Strength Results

Of the 54 studies included, 34 (63%) were excluded from the analysis for the reasons outlined in Figure 1. Twenty (37%) studies were included in the analysis, involving 26 treatment groups. For the studies included in the analysis, muscle strength changes are summarised for the relevant agonists (Table 3), antagonists (Table 4), and global muscle strength outcomes (Table 5). More detailed data, including treatment groups, mean, standard deviations, *p*-values, within-group differences, and follow-up time points, are outlined in Appendix A.

There were five (9%) studies that had a subset of their results included in the analysis [30,46,75,82,90]. Only the results from these studies where each participant received an injection to the same muscle group were included. The strength results from other muscle groups where not all participants received an injection were excluded [30,46,75,82,90]. This ensured that only muscle groups that received BoNT-A were included in the results. In four instances, the authors utilised the same data from a single cohort in multiple publications that were both included in the review [52,53,58,59,64,65,68,69].

#### 2.4.1. Agonist Muscle Strength—Upper Limb (Table 3)

Overall, few studies assessed upper limb agonists at the elbow, wrist, or fingers at all time points. No agonist or antagonist strength results were reported for the shoulder or forearm pronator or supinator muscles. Six (11%) studies included in the analysis assessed the upper limb agonist strength before and after BoNT-A injections [30,68,69,75,82,90]. The elbow, wrist, and finger flexors were the most frequently measured, typically in the early post-injection phase (0–6 weeks). Findings from seven treatment groups (from six studies) were assessed for upper limb agonists, with five out of the seven groups demonstrating no significant change in muscle strength [30,68,69,75,82,90].
toxins-16-00347-t003_Table 3Table 3Agonist muscle strength results from studies included in the analysis (*n* = 13).UL SPECIFIC MUSCLE GROUPS**Outcome Measure****Joint****Agonist****≤6/52****>6/52–≤3/12****>3/12–≤6/12****Stronger****NS****Weaker****Stronger****NS****Weaker****Stronger****NS****Weaker**Isometric (Nm)ElbowFlexor
[30] ^N^[69]



[68] 
MRC (0–5)
[90] ^§^[90] ^∞^






MRC (0–5)WristFlexor
[90] ^§^[90] ^∞^





[82]MRC (0–5)FingerFlexor
[75] *






LL SPECIFIC MUSCLE GROUPS
MRC (0–5)HipAdductor



[48]



Isokinetic—Peak torque 60°/s (Nm)KneeExtensors

[19] ª^∆Ω^





MVC—Isometric torque 40°/60° (Nm)

[33]





MVC—Concentric torque 30°/s/60°/s/90°/s (Nm)

[33]





MVC—Isokinetic—Concentric 60° (Nm)AnklePlantarflexors

[32] ^C^[32] ^E^
[32] ^E^[32] ^C^


QMA (kg)






[70]
MRC (0–5)

[70]
[48][46]

[70][46][81]
§—Subacute group; ∞—Chronic group; ª—*p*-value set at *p* < 0.01; * Tested at multiple time points in this range with both results NS; C—Control; E—Experimental; HHD—Handheld Dynamometer; ∆—Hip flexion 0°; Ω—Hip flexion 90°; kg—Kilograms; MRC—Medical Council Research Scale (0–5); MVC—Maximal Voluntary Contraction; N—Newtons; Nm—Newton Meters; NS—Not Significant; QMA—Quantitative Muscle Assessment—Fixed Myometry Muscle Testing. Significance is reported as *p* < 0.05 unless otherwise stated.


#### 2.4.2. Agonist Muscle Strength—Lower Limb (Table 3)

Seven (13%) studies involving eight treatment groups were included in the analysis examining the lower limb agonists’ muscle strength [19,32,33,46,48,70,81]. Of these studies, the knee extensors and ankle plantarflexors were the most frequently assessed muscle groups.

In the early phase (0–6 weeks) after BoNT-A injection, there were statistically significant decreases in knee extensor strength in two studies that utilised multiple testing positions [19,33]. No knee extensor muscle strength assessments were reported >6 weeks post-BoNT-A injection.

Regarding the plantarflexors, early post-injection (0–6 weeks) results indicate significant weakening [32,70]; however, between 6 weeks and 6 months, the majority of strength results demonstrated no significant change [46,70,81].

In summary, knee extensors and ankle plantarflexors appeared to weaken early post-injection (0–6 weeks). After this time, strength assessments were either not reported, or results were non-significantly different from baseline.

#### 2.4.3. Antagonist Muscle Strength—Upper Limb (Table 4)

Four studies (five treatment groups) reported antagonist upper limb strength with mixed results [30,75,82,90]. The subacute treatment group from Lim et al. (2016) found that the elbow extensors strengthened after injections to the flexors between 0 to 6 weeks post-injection [90]. All other antagonist results demonstrated no significant change between 0 to 6 weeks [30,75,90]. Further, the wrist and finger extensors were found to increase in strength between 3 to 6 months post-injection [82]. Notably, no data for upper limb antagonist groups were reported between 6 and 12 weeks post-injection.
toxins-16-00347-t004_Table 4Table 4Antagonist muscle strength results from studies included in the analysis (*n* = 11).UL SPECIFIC MUSCLE GROUPS
**Outcome Measure****Joint****Antagonist****≤6/52****>6/52–≤3/12****>3–≤6/12****Stronger****NS****Weaker****Stronger****NS****Weaker****Stronger****NS****Weaker**MVC—Isometric (Nm)ElbowExtensor
[30] ^N^






MRC (0–5)[90] ^§^[90] ^∞^






MRC (0–5)WristExtensor
[90] ^§^[90] ^∞^



[82]

MRC (0–5)FingerExtensor
[75] *



[82]

LL SPECIFIC MUSCLE GROUPS
MVC—Isometric torque 40°/60° (Nm)KneeFlexor[33]







MVC—Concentric torque 30°/s 60°/s (Nm)[33]







MVC—Concentric torque 90°/s (Nm)
[33]






MVC—Isokinetic torque60° (Nm)AnkleDorsiflexors[32] ^E^[32] ^C^

[32] ^E^[32] ^C^



MRC (0–5)
[45] ^G1^*[45] ^G2^*[54] ^E^*†[54] ^C^*†[47] ^Ta #^[47] ^Ca #^[47] ^St #^

[54] ^E^ †[54] ^C^ †[45] ^G1^[45] ^G2^[46][47] ^Ta #^[47] ^Ca #^[47] ^St #^

[83] ^R, L^[46]
^§^—Subacute group; ^∞^—Chronic group; *—Tested at multiple time points within range with both results NS, ^#^—*p* value set at *p* < 0.02 rather than *p* < 0.05; ^†^—measured at time post-injection (i.e., 5 days post-injection casting occurred + ∑ 27 days of casting); ^C^—Control, ^E^—Experimental; ^G1^—Group 1; ^G2^—Group 2; ^Ta^—Taping group, ^Ca^—Casting group; ^St^—Stretching group; MRC—Medical Council Research scale; MVC—Maximal Voluntary Contraction; N – Newtons; Nm – Newton Meters; NS—Not Significant; ^R^—Right leg; ^L^—Left leg; LL—Lower Limb; UL—Upper Limb. Significance was reported as *p* < 0.05 unless otherwise stated.


#### 2.4.4. Antagonist Muscle Strength—Lower Limb (Table 4)

Seven studies (12 treatment groups) examined the antagonist muscle strength post-injection to the agonists in the lower limb [32,33,45,46,47,54,83]. Predominantly, results for the effect of BoNT-A injection on lower limb antagonist strength were non-significant. Notably, Hameau et al. (2014) found that knee flexor strength increased in four out of five tests of the same treatment group using isokinetic dynamometry [33], and Cinone et al. (2009) found increases in dorsiflexion strength in both the experimental and control treatment groups between 0 and 6 weeks post-injection [32]. However, all other antagonist strength results demonstrated no significant change [45,46,47,54,75,83,90].

#### 2.4.5. Global Muscle Strength—Upper and Lower Limb (Table 5)

Grip strength was the most common global measure of upper limb strength, with the results across all time points demonstrating no significant change [30,52,53,75]. One study that used the MI as a global measure of strength for the upper limb indicated that the strength increased after BoNT-A injections in both treatment groups (Group 1: BoNT-A, Lycra sleeve and rehabilitation; Group 2: BoNT-A and rehabilitation only) [44]. Additionally, one study used the MI as a global strength measure for the lower limb, and the results from both treatment groups (BoNT-A and four weeks of isokinetic training; BoNT-A alone) demonstrated no significant strength changes after BoNT-A injections [32]. The results indicated that the effect of BoNT-A injections on global muscle strength in the upper and lower limbs was non-significant.toxins-16-00347-t005_Table 5Table 5Global muscle strength outcome measures from articles included in the analysis (*n* = 6).MovementOutcome Measure≤6/52>6/52–≤3/12>3/12–≤6/12>6/12StrongerNSWeakerStrongerNSWeakerStrongerNSWeakerStrongerNSWeakerGlobal ULMI[44] ^G1^[44] ^G2^

[44] ^G1^[44] ^G2^







Grip StrengthGSM (N)
[30]









HHD (kg)
[75] *

[53] ^E^[53] ^C^




[52] ^E^[52] ^C^
Global LLMI
[32] ^E^[32] ^C^

[32] ^E^[32] ^C^






* Tested at multiple time points within this range, both results NS; C—Control group; DCD—Dynamic Computerized Dynamometry; E—Experimental group; HHD—Handheld dynamometer; G1—Group 1; G2—Group 2; GSM—Grip Strength Meter; kg—Kilograms; LL—Lower Limb; MI—Motricity Index; N—Newtons; NS—Not Significant; UL—Upper limb. Significance is reported as *p* < 0.05 unless otherwise stated.


### 2.5. Meta-Analysis

Due to the heterogeneity of study designs, strength outcome measures, types of muscle contraction assessed, and assessment timeframes, a meta-analysis was not warranted [93]. When considering all the upper- and lower-limb agonists (Table 3), there was only one instance of more than one study reporting a strength outcome for the same muscle group using the same assessment tool at the same time point.

### 2.6. Spasticity Assessment Outcomes

Spasticity assessment outcomes were extracted for the studies included in the analysis. Outcome measures, means, standard deviations, *p*-values, and within-group changes are provided in Appendix A, where reported.

### 2.7. BoNT-A and Adjunctive Therapies

The type, dose, dilution, and units per muscle group of BoNT-A intramuscular injections administered to the participants are outlined in Appendix A. The adjunctive therapies for each study are also outlined in Appendix A where applicable.

### 2.8. Quality Assessment

The quality appraisal scores are presented in Table 1 and Table 2, and the results of the modified Downs and Black checklist subscales are reported in Appendix A. The median total score was 17.5 (range: 12–26) out of a maximum possible score of 28 on the modified Downs and Black checklist for all included studies, indicating fair-quality evidence [94]. Most studies satisfied the reporting criteria by outlining the hypothesis, main outcomes, patient characteristics, and interventions, with a median score of 10 (range: 8–11) out of a maximum score of 11 for this checklist section. External validity scores were poor, averaging 0.5 (range: 0–3) out of a possible 3. The median internal validity for confounding and bias subscale scores were 5 (range: 1–7) out of a possible 7 and 2.5 (range: 0–6) out of a possible 6, respectively. Only four studies reported power calculations [44,57,58,84].

Thirty-two (59%) studies included in this review were non-RCTs, which are susceptible to selection bias, confounding bias, performance bias, and observer bias [19,30,33,63,64,65,66,67,68,69,70,71,72,73,74,75,76,77,78,79,80,81,82,83,84,85,86,87,88,89,90,91]. Twelve (22%) studies were at risk of reporting bias due to missing data [5,19,30,48,51,52,53,55,57,70,72,74]. Nine (17%) of the included studies acknowledged their limitation of a small sample size [19,45,50,55,61,68,69,70,89]. Two (4%) studies failed to provide clear follow-up timeframes, creating difficulty when evaluating whether the change in strength post-BoNT-A was short-, medium-, or long-term [84,85].

The median PEDro score of the 22 RCTs included was 8 (range: 6–10) out of a possible 10, indicating good-quality evidence [95]. Randomised controlled trials often failed the criteria for blinding the participants, assessors, and therapists (criterion 5–7). Therapist blinding only occurred in 8/22 (36%) of the included RCTs.

## 3. Discussion

Overall, no statistically significant differences in muscle strength existed in ≥74% of all the injected muscles (i.e., agonist) and ≥64% of all the opposing muscles (i.e., antagonist). However, where significant changes in strength were reported, there was a tendency in the lower limbs for the agonist muscles to weaken [19,32,33,70], whereas the antagonist muscles did demonstrate some strength gains in both the upper and lower limbs [32,33]. No meta-analysis could be conducted due to the heterogeneous nature of the study designs, muscles injected, strength outcome measures, types of muscle contractions, and assessment time points [93]. This heterogeneity is a major barrier to the advancement of this field. The development of a ‘core set’ of outcome measures, such as that developed for neurological rehabilitation [96] or stroke [97], will enable future meta-analyses to report treatment effects and overall trends.

In a third of the studies (33%), strength outcomes were assessed using a global strength measure that was not specific to the injected muscle [44,52,53,58,59,62,63,64,65,72,76,77,79,80,87,88,89,91]. These types of strength assessment measures demonstrate low discriminative ability as they do not distinguish between individual muscles. From these results, we could not determine the direct effect of BoNT-A on the strength of the muscle injected. However, these results do indicate that the overall effect that BoNT-A has on global muscle strength may be negligible. Furthermore, in 11 (20%) studies, the wrist, elbow, and pronator muscles were injected and studies used a grip strength measure [30,52,53,57,58,59,61,75,86,88,89], but the action of grip primarily depends on the strength of the finger and thumb flexor muscles [98]. Since the application of the strength test was not specific to the muscle injected, this limits our understanding of the direct cause-and-effect relationship between BoNT-A injection and muscle strength. Therefore, strength testing assessments that are specific to the injected muscle are required to determine the effect of BoNT-A on the injected muscle.

The strength outcome measures reported in the systematic review have variable inter-rater reliability in neurological conditions [99,100,101,102]. Of the included studies, 36 (67%) used the MRC and/or the MI, which utilises traditional manual muscle testing to assess strength. Despite previous reports indicating very good inter-rater reliability of these scales [99,103,104], limitations exist [33]. The MRC does not quantify strength as a unit of force; rather, it is rated on a 6-point scale [104]. Manual muscle testing scales are prone to subjectivity and have reduced discriminability between moderate weakness and normal strength [105,106,107]. For example, reports have shown that 96% of strength results for elbow flexor strength were rated as a grade 4 [108]. Isokinetic dynamometry is the gold standard for muscle strength testing [109,110,111]. Although it was used in three (6%) studies in the lower limbs, it was not used in the upper limbs. This review highlights the need for standardised, accurate, and reliable muscle strength measurements to be used in future studies.

Botulinum neurotoxin-A has a therapeutic life of approximately three months. Studies have predominantly reported muscle strength in the first 0–6 weeks post-injection, with very few studies reporting strength changes >12 weeks and no studies reporting strength ≥12 months post-BoNT-A injection. No studies have reported upper limb strength (agonist or antagonist) between 6 and 12 weeks, which may be when the BoNT-A is most active. This does not align with current clinical practice guidelines, which recommend assessing outcomes at regular intervals (6 weeks, 3 months, 6 months, and >6 months) [7,112]. This systematic review highlights that the long-term implications of BoNT-A injections on muscle strength remain undefined and require further investigation.

It is widely known that BoNT-A injection reduces spasticity [14,23,25], and our findings (Appendix A) support this claim. The relationship between reducing the spastic response of the injected muscle and enabling optimal motor performance of the antagonist is clinically important [113]. For example, injection of the gastrocnemius muscle may enhance the training of the dorsiflexors to enable foot clearance in swing, and injection of the elbow flexor muscles may enable elbow extension for functional reach to grasp [114]. Simultaneously, it is also important not to ‘weaken’ the ankle plantarflexors to compromise ankle power generation at push-off or reduce the strength of the elbow flexors required for hand-to-face personal care. Evidence in rats suggests that BoNT-A weakens the injected muscle (i.e., agonist) [11,12,26,27,28]. In humans, the relationship between BoNT-A injection and changes in muscle strength is less clear. The heterogeneity of study designs, strength outcome measures, types of muscle contraction assessed, and assessment timeframes means that clinicians cannot be certain what the effect of BoNT-A on muscle strength will be. Furthermore, although the purpose of this review was to examine the relationship between BoNT-A and muscle strength, the impact on functional outcomes is unknown. The impact of BoNT-A on function, which is likely more meaningful to clinicians, has received less attention.

### Limitations

Other factors such as strength training, BoNT-A dosage for each muscle injection, and the severity and chronicity of spasticity may have influenced the muscle strength outcomes. Botulinum neurotoxin-A injections are often delivered as part of a package of care, which may include strength training, splinting, or casting. All these therapies may potentially impact muscle strength. Examining factors such as the use of adjunctive therapies and the type and dose of BoNT-A injection was beyond the scope of this review. However, this data is presented in Appendix A for transparency.

Furthermore, there were varying degrees of reporting of the use of concomitant medications. Thirty-one (57%) studies did not stipulate whether participants were also taking any other medications, nineteen (35%) studies reported that participants did have anti-spastic medications during the period of BoNT-A treatment [30,46,48,50,54,55,56,57,58,59,67,73,74,75,76,81,82,83,84], and only four (7%) studies reported that no anti-spastic medications were taken in addition to the BoNT-A injections as per the studies’ protocols [4,47,80,89].

A significant proportion of the studies involved small sample sizes, which may have increased the variability in the results and the risk of publication bias [115]. Fifty (93%) studies did not report sample size calculations, and given the predominance of small cohorts, it is likely that most studies were underpowered. Therefore, these studies may not accurately detect the effect of BoNT-A injections on muscle strength, so caution is recommended when interpreting the results.

While this review reports the current research examining the impact of BoNT-A on muscle strength, it does not consider the effect that any changes in spasticity and muscle strength may have on upper or lower limb function.

Finally, we did not include studies that were not published in English, which may have resulted in relevant studies being overlooked and potentially important data being omitted from the review.

## 4. Conclusions

Overall, the impact of BoNT-A injections on muscle strength, especially when examining long-term outcomes, remains uncertain. The muscle strength outcome measures used were often subjective, were not specific to the muscle injected, and may not have the capacity to accurately detect change. The findings of this review suggest that further research is warranted to systematically investigate the impact that BoNT-A injections have on muscle strength.

## 5. Future Directions

In order to better understand the impact that BoNT-A injections have on muscle strength, there is a need to conduct adequately powered studies investigating people with a range of neurological conditions using objective strength testing. These assessments must be able to differentiate between muscle groups and should be repeated at short-, medium-, and long-term time points following BoNT-A administration. Future studies should also explore the effect of changes in muscle strength on functional outcomes, as this may be of greater clinical significance.

## 6. Methods

### 6.1. Review of the Literature

This systematic review was registered with the International Prospective Register of Systematic Reviews (PROSPERO) database, registration number CRD42022315241. To enable a systematic and non-biased approach, the methodology and reporting of results used throughout this review followed the PRISMA guidelines and checklist [43].

### 6.2. Search Strategy

A systematic literature search was conducted in March 2024. The following eight databases were searched for relevant articles: MEDLINE, CINAHL, Embase, Cochrane Central Register of Controlled Trials, Web of Science, PubMed, Physiotherapy Evidence Database (PEDro), and Google Scholar, and the search was limited to English language and human studies. Where possible, the search was performed using relevant medical subject headings (MeSH) and keywords mapped to the titles and abstracts of articles. Wildcard and truncation symbols were used to capture all suffix variations of a root word. All databases were searched since inception, and search strategies were customised for each database. The search strategy for all databases is outlined in Appendix A. The first 200 references using relevant search terms were exported for screening for Google Scholar, as Haddaway et al. (2015) recommended [116].

### 6.3. Eligibility Criteria

Studies were included in the review if they met the following criteria: (a) included participants with an adult-onset (i.e., ≥18 years of age) neurological condition; (b) assessed focal muscle spasticity in any upper or lower-limb muscle group; (c) participants received BoNT-A intramuscular injections; (d) included a clinical (e.g., Medical Research Council scale), instrumented (e.g., hand-held dynamometry) or laboratory-based (e.g., isokinetic dynamometry) measure of strength pre and post-injection; and (e) were randomised controlled trials (RCTs), non-RCTs, cohort studies, or reports with *n* ≥ 10. Studies were excluded if they met any of the following criteria: (a) participants had complete spinal cord injuries, (b) systematic reviews, (c) grey literature, (d) published in a language other than English, or (e) animal-based studies.

### 6.4. Selection of Articles

All articles from the searches were added to Endnote for collation and then exported into Covidence screening software (Veritas Health Innovation, Melbourne, Australia, available at www.covidence.org) to remove duplicates. The screening process was then completed. Two independent review team members (R.G, Z.Y, P.M.M, C.C.A.W) screened all titles and abstracts to identify potential articles based on the inclusion criteria. Where the subject title and abstract were inconclusive, the full text of the article was appraised. After articles passed the initial screening, full-text articles were obtained to determine inclusion or exclusion. Where only an abstract was found or full-text articles could not be obtained, the article was excluded. Two review team members (R.G, Z.Y, P.M.M, C.C.A.W) independently reviewed each full-text article to determine whether it met the inclusion criteria. Both reviewers agreed to either include or exclude articles based on the inclusion criteria. A third review team member (M.B or G.W) resolved disagreements between judgments. Bibliographic reference lists of all included articles and systematic reviews were screened manually to identify additional literature not recognised through the electronic literature database searches.

### 6.5. Data Extraction

One author (R.G) extracted relevant data from included articles and recorded data using a Microsoft Excel (Version 16.77.1) template and customised Microsoft Word (Version 16.77.1) tables, which were checked for accuracy and completeness by a second author (Y.Z, M.B, G.W). The following data were extracted from each of the included studies: study details, participant demographics, strength outcome measurement data (inclusive of type of measure, muscles assessed, and time points of assessment), and spasticity data (inclusive of BoNT-A administration and dosing and spasticity assessment outcomes).

Where data were missing or not reported, authors were contacted via email twice and given at least two months to respond to provide further data.

### 6.6. Methodological Quality Assessment

Two review team members (R.G, Y.Z, P.M.M, C.C.A.W) assessed the methodological quality of all included studies, and a third reviewer (MB) resolved any discrepancies. The modified Downs and Black checklist [117] (27-item scale) was used for all included studies, and the PEDro scale was used for the RCTs [118,119,120]. We used two different tools to assess the studies, as they differed in the assessment criteria and could be applied to different study designs. The PEDro scale is only applicable for assessing RCTs, and it also assesses the blinding of therapists [121]. The modified Downs and Black checklist was selected because it can be applied to non-RCT studies [118,121]. These scales have been reported to have moderate comparability [118].

#### 6.6.1. The Modified Downs and Black Checklist

The modified Downs and Black checklist assesses five domains [117]. These include reporting (9 items), external validity (3 items), internal validity—bias (7 items), internal validity—confounding (6 items), and power (1 item). For this version of the checklist, we changed the scoring of item 27, where instead of rating the power within a certain range, we checked whether a power analysis was performed or not. As a result, the highest score for item 27 was 1 (if a power analysis was conducted), and the highest score for the checklist was 28 (instead of 32) [94]. This method has been utilized in previous studies [118,122]. The modified Downs and Black checklist scores are interpreted as excellent (26–28), good (20–25), fair (15–19), or poor (≤14) [94].

#### 6.6.2. The PEDro Scale

The PEDro scale is an 11-item scale designed to examine the internal and external validity of RCTs and provide a standardised format for quality evaluation [123]. The total score is out of a possible 10 and is derived from the number of criteria satisfied for each study. The first criterion, ‘eligibility criteria,’ is not calculated in the final score [119].

### 6.7. Data Analysis

Strength outcomes were analysed according to agonist muscle strength (i.e., the injected muscle), antagonist muscle strength (i.e., the opposing muscle), and global muscle strength. Studies were included in one or more of these analyses if they met the following criteria: (1) all participants underwent a strength assessment of either the injected muscle and/or the opposing muscle group or underwent a relevant global strength assessment; (2) muscle strength was measured before and at least once after injection; (3) all participants needed to have the same relevant muscle group injected, and if only a proportion of the cohort had the muscle group injected, the study was excluded from the analysis; (4) all mean and standard deviation data were reported or could be calculated; and (5) in studies where participants underwent repeated injections, only the data from baseline to the first follow-up (i.e., before a second injection) could be included in the analysis. For example, if the strength of a specific muscle (e.g., the ankle plantarflexors) was reported, all participants were required to have been injected with BoNT-A in this muscle group to be included in the agonist analysis. For the antagonists (e.g., the ankle dorsiflexors), where strength was reported, all participants must have been injected with BoNT-A in the opposing muscle group (e.g., the plantarflexors). Further, where muscle strength was reported using a global measure (i.e., grip strength), all participants were required to have had the same relevant muscle group injected (e.g., wrist or finger flexors) for this strength result to be included. If only a proportion of participants had the same muscle group injected, but the muscle strength result was reported for the entire cohort, those results were excluded from further analysis.

Where data were presented in such a way that they could not be pooled for analysis, two attempts were made to contact the authors, who were given a minimum of two months to respond. In cases where data were not provided, the study was excluded from the analysis.

Once the relevant agonist, antagonist, and global strength data were extracted, the results were categorised according to the upper and lower limb, joint region, and strength outcome measures used. Significant changes in strength, the direction of change (i.e., improvement or worsening), and non-significant (NS) findings were extracted. The strength results from each study were then categorised according to spasticity guideline-recommended follow-up timeframes (i.e., 0 to 6 weeks, 6 to 12 weeks, 3 to 6 months, and greater than 6 months) [112].

In studies which had multiple treatment arms, data were extracted for each treatment arm that received BoNT-A injections. Thus, in some instances, numerous results from the same study assessing different treatment arms were extracted for analysis.

Statistical significance was defined as *p* ≤ 0.05 unless otherwise stated by the study. In instances where the *p*-value was set differently, this was indicated with a symbol and noted in the table caption.

## Figures and Tables

**Figure 1 toxins-16-00347-f001:**
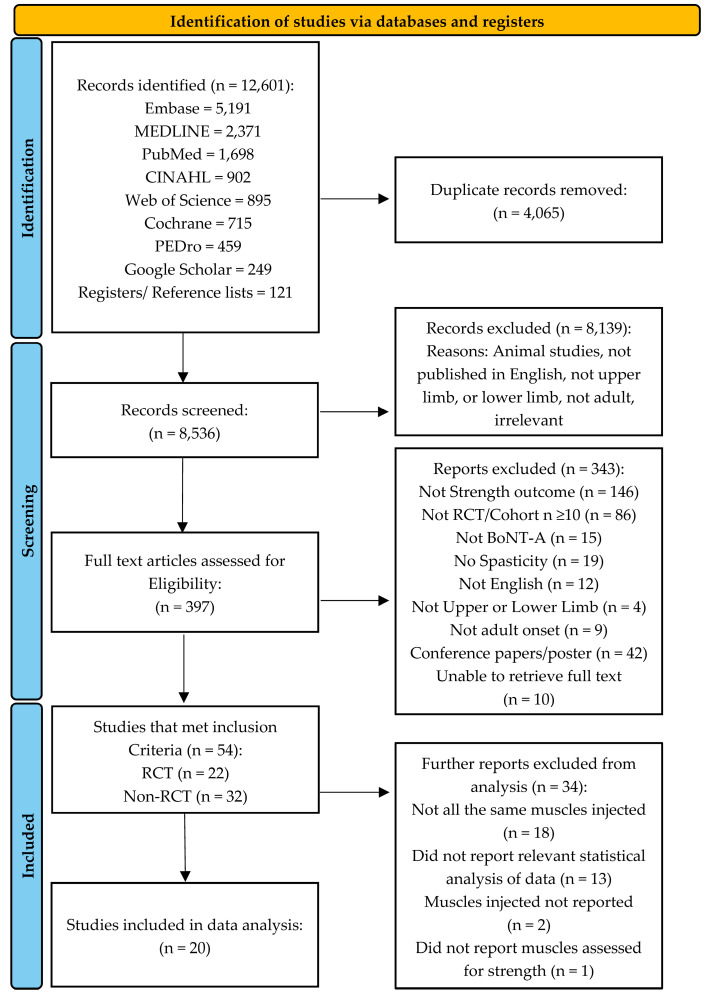
Flow diagram of study identification to obtain articles for review inclusion.

**Table 1 toxins-16-00347-t001:** Study characteristics of the included studies describing injection of upper limb and strength testing (*n* = 38).

Author Study Design	Population	Sample (*n*)	Mean Age (y)	Muscles Injected	Outcome Measures	Follow-up Time Points	mD&B	PEDro
Baguley 2022 [63] Non-RCT	Stroke/TBI	30	G1:60 G2:58	PM, SC, BRA, BB, BR, PT, PQ, FCR, FCU FDS, FDP, hand intrinsics, AP, FPL, FPB, AbDM	DCD grasp	4/52	14	
Barden 2014 [64] Non-RCT	TBI/Stroke	28	51	PM, SC, BRA, BR, BB, PT/PQ, FCR, FCU, FDS, FDP, FPL, AP, FPB, LUM interossei, AbDM	DCD pincer pinch max, DCD lat pinch max	4/52	16	
Barden 2014 [65] Non-RCT	TBI/Stroke	28	51	UL #	DCD grip max	4/52	15	
Bhakta 2000 [5] RCT	Stroke	40	E:60 C:54	BB, BR FCU, FDS, FDP	MRC Grip	2/52, 6/52 12/52	22	10
Bumbea 2023 [66] Non-RCT	Stroke	160	63	D, P, PT FDP, FDS, FCR FPL	MRC	3/12	19	
Caty 2009 [67] Non-RCT	Stroke	20	59	BB, BRA, PT, FCR, FDP, FPL	Grip-HHD Key pinch gauge	2/12	12	
Chen 2020 [69] Non-RCT	Stroke	10	52	BB	MVC	3/52	15	
Chen 2022 [68] Non-RCT	Stroke	12	52	BB	MVC	3/12	16	
Fawzi 2023 [71] Non-RCT	Stroke	50	48	BB^ FCR^	MRC	3–4/52	15	
Franck 2021 [72] Non-RCT	Stroke	10	56	BB, PT FDP, FDS, FPL FCR, FCU	Grip-HHD MI	1/52 3–6/52 3/12	20	
Gandolfi 2019 [49] RCT	Stroke	32	E:59 C:59	BB, BRA, BR, PM FCR, FCU, FDP, FDL, FDS, FPL, FPB, LUM, OP, ECRB, ECUL^	MRC	5/52	22	8
Giray 2020 [44] RCT	Stroke	20	46	BB, BRA, PT, PQ FCR, FCU, FDS, FDP, FPL	MI	3/52 3/12	21	7
Gracies 2009 [50] RCT	Stroke/TBI	21	G1:46 G2:52 G3:47	BB	MVP isometric HHD	1/12	22	8
Intiso 2014 [73] Non-RCT	TBI/CP	BI (*n* = 16) CP (*n* = 6)	38	PM, BB, BR, PT, FDS FDP, FCU, FCR, FPL	MRC	4/52 4/12	17	
Kulkarni 2004 [74] Non-RCT	CVA, MS, CP, BI, Other	72	45	UL #	MRC	4–6/52	14	
Lannin 2020 [53] RCT	Stroke	139	61	FCR, FCU, FDS, FDP, FPL, ED, EDM, PL ECRL, ECU, ECRB	Grip-HHD	3/12	20	8
Lannin 2022 [52] RCT	Stroke	140	61	FCR, FCU, FDS, FDP, FPL, ED, EDM, PL ECRL, ECU, ECRB	Grip-HHD	12/12	21	8
Lee 2018 [75] Non-RCT	Stroke	15	45	D, BB, BRA, BR FCU, FDP, FDS, AP, PT, FPB, FPL, OP, LUM	MRC Grip—HHD	2/52 6/52	13	
Lim 2016 [90] Non-RCT	Stroke	18	SA:63 Ch:52	BB, BR, BRA FCR, FCU FDP, FDS, FPL	MRC	4/52	16	
Macher 2021 [55] RCT	Stroke (*n* = 10) ISCI (*n* = 1)	11	68 M	BB, BR, PT, Sup, FDS, FDP, FPB, AP, FPL, OP, FCR, FCU	HHD—EF	1/52, 4/52 8/52, 3–7/12	22	7
Marque 2019 [77] Non-RCT	Stroke	330	54	D, SC, PM, FDS, FDP, BB, BRA, BR, PL, PT, PQ, FCR, FPL, FCU, Other	MI	4/52 3/12 12/12	16	
Meythaler 2009 [57] RCT	Stroke	21	53	WF, EF, PT	MRC Grip-HHD Lat Pinch Dynamometer	12/52 24/52	23 ^	8
Miscio 2004 [78] Non-RCT	Stroke	18	48	PT, FCR, FCU FDS, FDP, PL, FPL, OP	Isometric strain gauge	2/52, 1/12 2/12, 3/12	17	
Pandyan 2002 [30] Non-RCT	Stroke	14	57 ^	BB, BR FDL (FDP)	Isometric force transducer Grip—GSM	4/52	14 ^	
Paolucci 2021 [79] Non-RCT	Stroke	44	E:66 C:65	SC, BB-med, BB-lat, BRA, BR, PT, FCR, FCU FDS, FDP, 1st FF	MI	6/52 3/12	22	
Picelli 2021 [80] Non-RCT	Stroke	83	64	Sh Add, EE, FF EF, TF, WF, Pron	MI	4/52 12/52 24/52	18	
Reiter 1996 [91] Non-RCT	Stroke/TBI	17	58	BB, FCU, FCR, FDP, FDS, FPL	MI MRC	1/52 1/12, 2/12, 3/12, 4,12, 5/12, 6/12	16	
Rousseaux 2002 [82] Non-RCT	Stroke	20	54	PM, D (anterior) EF, PT, PL, WF, FDS, FPL	MRC	2/52, 2/12 5/12	16	
Sarzynska-Dlugosz 2020 [84] Non-RCT	Stroke	57	57	UL #	MRC	~4/12	16	
Shaw 2010 [58] RCT	Stroke	333	67	PM, BB, BR, PT, FDS, FDP, FPL Forearm flexors, FCU, FCR	Grip-HHD MI	1/12	26	8
Shaw 2011 [59] RCT	Stroke	333	67	PM, BB, BR, PT, FDS, FDP, FPL Forearm flexors, FCU, FCR	Grip-HHD MI	1/12	21	7
Simpson 1996 [62] RCT	Stroke	37	59	BB, FCR, FCU	Grip	2/52, 6/52 10/52, 4/12	19	8
Slawek 2005 [86] Non-RCT	Stroke	21	52	PM, BB, BR, PT, FCU, FCR, FDS, FDP, FPL, AP	MRC	2/52, 4/52, 6/52, 10/52, 16/52	12	
Tsuchiya 2016 [87] Non-RCT	Stroke (*n* = 14) ISCI (*n* = 1)	15	52	BB, PT, FCR, FCU FDS, FPL, AP	Grip-HHD	10/7 4/12	19	
Turcu-Stiolica 2021 [60] RCT	Stroke	34	E:60 C:61	UL #	MRC-UL	6/12	17	6
Wallace 2020 [61] RCT	Stroke	28	49	FCR, FCU, PT FDS, FDP, FPL, LUM	Grip-HHD Isometric Servomotor	5/52	22	9
Wang 2002 [88] Non-RCT	Stroke	16	62	BB, BR, FCU, FCR FDS, FDP, FPL, VI	Grip-HHD Pinch gauge	2/52, 4/52 8/52, 3/12	16	
Woldag 2003 [89] Non-RCT	Stroke	10	45	FCR, FCU, FDP, FDS^	Grip-Max digital multimyometer	4/52, 8/52 12/52	17	

^—Data supplied by author upon request; #—did not specify which muscles; AbDM—Abductor Digiti Minimi; AP—Adductor pollicis; BB—Biceps Brachii; BI—Brain injury; BRA—Brachialis; BR—Brachioradialis; C—Control; Ch—Chronic; CP—Cerebral Palsy; CVA—Cerebrovascular accident; D—Deltoid; DCD—Dynamic Computerised Dynamometry; E—Experimental; ECRB—Extensor Carpi Radialis Brevis; ECRL—Extensor Carpi Radialis Longus; ECUL—Extensor Carpi Ulnaris Longus; ED—Extensor Digitorum; EDM—Extensor Digiti Minimi; EE—Elbow Extension; EF—Elbow Flexion; FCR—Flexor Carpi Radialis; FCU—Flexor Carpi Ulnaris; FDP-Flexor Digitorum Profundus; FDS—Flexor Digitorum Superficialis; FDL—Flexor Digitorum Longus; FE—Finger Extension; FF—Finger Flexion; FPB—Flexor Pollicis Brevis; FPL—Flexor Pollicis Longus; G—Group; GSM—Grip Strength Meter; HHD—Handheld Dynamometry; inj—injection; ISCI—Incomplete Spinal Cord Injury; lat—lateral; LUM—Lumbricals; Max—Maximum; med—Medial; MI—Motricity Index; MRC—Medical Research Council scale; MS—Multiple Sclerosis; MVC—Maximal Voluntary Contraction; mD&B—Modified Downs and Black scale; MVP—Maximal Voluntary Power; *n*—Sample; OP—Opponens Pollicis; P—Pectoralis muscles; PEDro—Physiotherapy Evidence Database scale; PL—Palmaris Longus; PM—Pectoralis Major; Pron—Pronators; PT—Pronator Teres; PQ—Pronator Quadratus; RCT—Randomised Controlled Trial; SA—Subacute; Sh—Shoulder; SC—Subscapularis; Sup—Supinators; TF—Thumb Flexion; TBI—Traumatic Brain Injury; UL—Upper limb; VI—Volar Interossei; WF—Wrist Flexion; y—years.

**Table 2 toxins-16-00347-t002:** Study characteristics of the included studies describing injection of the lower limb and strength testing (*n* = 20).

Author Study Type	Population	Sample Size (*n*)	Mean Age (y)	Muscles Injected	Outcome Measure	Follow-Up Time Points	mD&B	PEDro
Baricich 2019 [45] RCT	Stroke	30	59	GN—lat GN—med, Sol	MRC	10/7 2.5/52, 3/12	20	8
Bernuz 2012 [19] Non-RCT	ISCI	15	43	RF	Isokinetic peak voluntary torque 60°/s	4–6/52	15	
Bollens 2013 [46] RCT	Stroke	16	52.3	Sol TP, FHL	MRC	2/12 6/12	22	7
Carda 2011 [47] RCT	Stroke	69	Ta:62 Ca:65 St:60	GN—med GN—lat Sol	MRC-DF	3/52 3/12	23	7
Cinone 2019 [32] RCT	Stroke	25	E:56 C:56	GN—med GN—lat Sol	MI Isokinetic dynamometer—Peak Torque 60°/s	5/52 8/52	19	8
de Niet 2015 [70] Non-RCT	HSP	25	E:48 C:46	GN—Med GN—Lat and Sol	MRC QMA	4/52 18/52	17	
Diniz de Lima 2021 [48] RCT	HSP	55	43	AM GN + Sol	MRC	8/52	19	8
Fawzi 2023 [71] Non-RCT	Stroke	50	48	GN/Sol ^	MRC	3–4/52	15	
Hameau 2014 [33] Non-RCT	Stroke	14	54	RF	Isokinetic Dynamometer MVC-peak torque	1/12	15	
Intiso 2014 [73] Non-RCT	BI/CP	14 (and 3 CP)	38	ADDLBM, RF, BF, GN—Med, GN—lat, Sol, TP, TA, FDL, FHL	MRC	4/52 4/12	17	
Kaji 2022 [51] RCT	Stroke	31	E:62 C:63	TP GN—Med	MRC	1/12 2/12	18	8
Kulkarni 2004 [74] Non-RCT	CVA, MS, CP, BI, other	72	45.3	UL # HS, Hip Abd	MRC	4–6/52	14	
Leung 2019 [54] RCT	ABI	10	E:27 £ C:39 £	GN, Sol +/− TP	MRC	2/52 post cast 8/52 post cast	22	8
López de Munain 2019 [76] Non-RCT	Stroke	100	58.2	LL #	MI	1/12 ± 7/7 3–5/12	19	
Mancini 2005 [56] RCT	Stroke	45	G1:62 £ G2:59 £ G3:60 £	GN—med, GN—lat Sol, TP, FDL, FDB, EHL, EH	MRC	4/52 4/12	20	9
Picelli 2021 [80] Non-RCT	Stroke	83	64	PF, Ankle invertors	MI	4/52, 3/12, 6/12	18	
Rousseaux 2005 [81] Non-RCT	Stroke	47	52	TP, TA, GN—med GN—lat, Sol, FDL, FHL	MRC	2–3/52 2–3/12, 5/12	15	
Rousseaux 2007 [83] Non-RCT	HSP	15	48 M	Hip Add Sol, TP	MRC	2–3/52 2–3/12, 5/12	17	
Servelhere 2018 [85] Non-RCT	HSP	33	42	Hip Add, QUAD, HS, GN, Sol, TP EHL, FDL, FDB, FHB, QL, TA	MRC	~1.5/12 ± ~2/52	15	
Yan 2018 [4] RCT	ISCI	336	BoNT-A: 37 No drug: 35 Baclofen: 37	Hip Add HS	mMRC (0–6) #	2/52 4/52 6/52	21	7

£—Age at injury; ^—Data supplied by author upon request; #—did not specify which muscles; ABI—Acquired Brain Injury; ADDLBM, adductor-longus-brevis-magnus; Add—Adductor/Adduction; Abd—Abduction; AM—Adductor Magnus; BI—Brain injury; BF—Biceps Femoris; BoNT-A—Botulinum neurotoxin-A; C—Control; Ca—Casting; CP—Cerebral Palsy; CVA—Cerebrovascular Accident; DF—Dorsiflexion; E—Experimental; EH—Extensor Hallucis; EHL—Extensor Hallucis Longus; FDB—Flexor Digitorum Brevis; FDL—Flexor Digitorum Longus; FHL—Flexor Hallucis Longus; FHB—Flexor Hallucis Brevis; GN—Gastrocnemius; HS—Hamstrings; HSP—Hereditary Spastic Paraplegia; ISCI—Incomplete Spinal cord injury; lat—Lateral; LL—Lower Limb; M—Median; Max—Maximum; Medial—med; MI—Motricity Index; mMRC—Modified Medical Research Council Scale; mD&B—Modified Downs and Black scale; MRC—Medical Research Council Scale; MS—Multiple Sclerosis; *n*—Sample; Sol—Soleus; St—Stretching; PEDro—Physiotherapy Evidence Database scale; PF—Plantarflexion; QMA—Quantitative Muscle Assessment; QL—Quadratus Lumborum; QUAD—Quadriceps; RCT—Randomised Controlled Trial; RF—Rectus Femoris; SKG—Stiff Knee Gait; Ta—Taping; TA—Tibialis Anterior; TP—Tibialis Posterior; y—years.

## Data Availability

All analysed data are reported in the Appendix A. The data presented in this study are available upon request from the corresponding author.

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
