# Peer review of "The Effect of Botulinum Neurotoxin-A (BoNT-A) on Muscle Strength in Adult-Onset Neurological Conditions with Focal Muscle Spasticity: A Systematic Review"

_toxins, 2024, doi:10.3390/toxins16080347_

Round 1

Reviewer 1 Report

Comments and Suggestions for Authors

It is a very good paper. Suggestions for improvement:

1. The results are clearly presented with well-organized tables and supplementary materials, but a more detailed narrative summary of the key findings within the text would aid in understanding.

2. The discussion effectively highlights the study's key findings and their implications but could benefit from a deeper exploration of clinical implications. 

Comments on the Quality of English Language

Minor editing of the English language is required to correct occasional typographical errors and improve sentence structure for better readability. Additionally, ensure consistent use of terminology, such as using "Botulinum neurotoxin-A" or "BoNT-A" uniformly throughout the text. 

Reviewer 2 Report

Comments and Suggestions for Authors

This systematic review appears, for the most part, well-structured and follows the PRISMA checklist. The main issues that I suggest addressing are to better state in the introduction why it is relevant to address this argument and to discuss the results critically.

Following the other suggestions:

- better structuring the abstract following the PRISMA checklist for the abstract.

- The introduction appears too long in comparison with the discussion

-The discussion is too short, in some parts appearing as a results summary without a critical interpretation of them and an attempt to give an answer on what has been highlighted by the results.

Reviewer 3 Report

Comments and Suggestions for Authors

This review is important regarding the conclusions that must be followed by a meta-analysis. You said "Overall, the impact of BoNT-A on muscle strength remains inconclusive." only based on some percentages. You included 21 studies in data analysis and no meta-analysis could be conducted as studies utilised different outcome measures to assess strength across a range of different muscle groups. I recommend to perform meta-analysis by muscle groups, even if there are 2 or 3 included studies, at least at 6 weeks post-BoNT-A injection. 

The PRISMA methodology is very well respected.

You are right the assessments were reported >6 weeks 231 post-BoNT-A injection and this is a very important conclusion.

Reviewer 4 Report

Comments and Suggestions for Authors

In this systemic review, authors described the effect of Botulinum neurotoxin-A (BoNT-A) on muscle strength in adult-onset neurological conditions with focal muscle spasticity. Majority of studies reported the short term effect within six week post injection and only few reports on long-term effect of BoNT-A injection. Its impact on the muscle strength with regards to the functional outcomes remains questionable.

 The manuscript is well-presented and, application of materials and methods  are reasonable for this review.

No more comments from this reviewers.
